# Maximum Common Subgraph Guided Graph Retrieval: Late and Early Interaction Networks

**Indradyumna Roy    Soumen Chakrabarti    Abir De**
Indian Institute of Technology Bombay
{indraroy15, soumen, abir}@cse.iitb.ac.in

## Abstract

The graph retrieval problem is to search in a large corpus of graphs for ones that are most similar to a query graph. A common consideration for scoring similarity is the maximum common subgraph (MCS) between the query and corpus graphs, usually counting the number of common edges (i.e., MCES). In some applications, it is also desirable that the common subgraph be connected, i.e., the maximum common connected subgraph (MCCS). Finding exact MCES and MCCS is intractable, but may be unnecessary if ranking corpus graphs by relevance is the goal. We design fast and trainable neural functions that approximate MCES and MCCS well. Late interaction methods compute representations for the query and corpus graphs separately, and compare these representations using simple similarity functions at the last stage, leading to highly scalable systems. Early interaction methods combine information from both graphs right from the input stages, are usually considerably more accurate, but slower. We propose both late and early interaction neural MCES and MCCS formulations. They are both based on a continuous relaxation of a node alignment matrix between query and corpus nodes. For MCCS, we propose a novel differentiable 'gossip' network for estimating the size of the largest connected common subgraph. Extensive experiments with seven data sets show that our proposals are superior among late interaction models in terms of both accuracy and speed. Our early interaction models provide accuracy competitive with the state of the art, at substantially greater speeds.

## 1  Introduction

Given a query graph, the graph retrieval problem is to search for *relevant* or *similar* response graphs from a corpus of graphs [1, 2, 3, 4, 5, 6, 7, 8, 9, 10, 11]. Depending on the application, the notion of relevance may involve graph edit distance (GED) [12, 13, 14], the size of the maximum common subgraph (MCS) [5, 6, 7, 8, 9], full graph or subgraph isomorphism [5, 10, 11], *etc*. In this work, we focus on two variations of MCS-based relevance measures: (i) maximum common edge subgraph (MCES) [15], which has applications in distributed computing [16, 15] and molecule search [17, 18, 19, 20] and (ii) maximum common connected subgraph (MCCS) [21], which has applications in keyword search over knowledge graphs [22, 23], software development [24, 25, 26], image analysis [27, 28, 29], *etc*.

In recent years, there has been an increasing interest in designing neural graph retrieval models [5, 6, 7, 8, 9, 10]. However, most of them learn black box relevance models which provide suboptimal performance in the context of MCS based retrieval (Section 4). Moreover, they do not provide intermediate matching evidence to justify their scores and therefore, they lack interpretability. In this context, Li et al. [5] proposed a graph matching network (GMN) [5] based on a cross-graph attention mechanism, which works extremely well in practice (Section 4). Nevertheless, it suffers from three key limitations, leaving considerable scope for the design of enhanced retrieval models. (i) Similar to other graph retrieval models, it uses a general-purpose scoring layer, which renders it suboptimal in the context of MCS based graph retrieval. (ii) As acknowledged by the authors, GMN is slow in both

training and inference, due to the presence of the expensive cross-attention mechanism. (iii) MCS or any graph similarity function entails an *injective* mapping between nodes and edges across the graph pairs. In contrast, cross-attention induces potentially inconsistent and non-injective mappings, where a given query node can be mapped to multiple corpus nodes and vice-versa.

## 1.1 Our contributions

We begin by writing down (the combinatorial forms of) MCES and MCCS objectives in specific ways that facilitate subsequent adaptation to neural optimization using both late and early interaction networks. Notably, these networks are trained end-to-end, using only the distant supervision by MCES or MCCS *values* of query-corpus graph pairs, without explicit annotation of the structural mappings identifying the underlying common subgraph.

**Late interaction models.** We use a graph neural network (GNN) to first compute the node embeddings of the query and corpus graphs *independently* of each other and then deploy an interaction network for computing the relevance scores. This decoupling between embedding computation and interaction steps leads to efficient training and inference. We introduce LMCES and LMCCS, two late interaction neural architectures for MCES and MCCS based graph retrieval respectively. The interaction model is a differentiable graph alignment planner. It learns a Gumbel-Sinkhorn (GS) network to provide an *approximate alignment plan* between the query and corpus graphs. In contrast to GMN [5], it induces an approximately injective mapping between the nodes and edges of the query and corpus graphs. The MCES objective is then computed as a differentiable network applied to this mapping. For MCCS, we further develop a novel *differentiable gossip network* that computes the size of the largest connected component in the common subgraph estimated from the above mapping. These neural gadgets may be of independent interest.

**Early interaction model.** In the early interaction model, we perform the interaction step during the node embedding computation phase, which makes the query and corpus embeddings dependent on each other. This improves predictive power, at the cost of additional training and inference time. Here, we propose XMCS (cross-MCS), an early interaction model that works well for both MCES and MCCS based graph retrieval. At each propagation layer of the GNN, we first refine the alignment plan using the embeddings computed in the previous layer, then update the underlying coverage objective using the refined alignment and finally use these signals to compute the node embeddings of the current layer.

**Comprehensive evaluation.** We experiment with seven diverse datasets, which show that: (i) our late and early interaction models outperform the corresponding state-of-the-art methods in terms of both accuracy and inference time; (ii) in many cases, LMCES and LMCCS outperform the early interaction model of GMN [5]; and (iii) GMN's accuracy can be significantly improved by substituting its final layer with our MCS-specific neural surrogate.

## 1.2 Related work

**Combinatorial algorithms for MCS.** Both MCES and MCCS are NP-complete [15]. Several works designed heuristics for computing MCES for specific types of graphs [18, 17, 30]. Bahiense et al. [15] formulated MCES as an integer programming problem, provided a polyhedral analysis of the underlying formulation, and finally designed a branch and cut algorithm to solve it. Such polyhedral study for MCES was also performed by others [31, 32]. Combinatorial methods for different variants of MCCS have also been thoroughly studied. Some of them provide exact MCCS [33, 34, 35]. McCreesh et al. [33] proposed McSplit, a branch and bound algorithm for maximum common induced and connected graph, which is efficient in terms of time and memory. Other works provide effective heuristics to find approximate MCCS [36, 37]. However, these methods are not differentiable and therefore not suitable for data-driven MCS estimation.

**Learning models for MCS.** There are some recent efforts to design machine learning models for graph similarity and search [38, 7]. Among them, Bai et al. [38, GLsearch] compute MCS between two graphs using a reinforcement learning setup. In contrast, we consider a supervised learning setting for graph retrieval. Although Bai et al. [7, GraphSim] focus on the supervised learning scenario, their relevance scoring model performs poorly for MCS based retrieval.

**Graph matching for computer vision.** Neural models for graph matching are used in applications of computer vision for computing image similarity, object detection, etc. However, these applications permit explicit supervision of the underlying node alignments [39, 40, 41, 42, 43, 44, 45, 46, 47, 48].

They adopt different types of losses which include permutation loss [41, 42, 43, 44], Hungarian loss [46], and displacement loss [40]. In our problem, we only use distant supervision in the form of size of the underlying MCS. Moreover, these work mostly consider graph matching problems, whereas we consider maximum common subgraph detection using distant supervision from MCS score alone.

**Graph representation learning.** Representation learning on graphs has been widely researched in the last decade [49, 50, 51, 52, 53, 54]. Among them, graph neural networks (GNN) are the most popular node embedding models [49, 50, 51, 52, 55, 56]. Given a node $u$, a GNN collects information from $K$-hop neighbors of the node $u$ and applies a symmetric aggregator on top of it to obtain the representation vector of the nodes. In the context of graph retrieval, the node embeddings are used in two ways. In the first approach, they are further aggregated into graph embeddings, which are then used to compute the similarity between query and corpus graphs, by comparing the embeddings in the vector space [10, 5]. The second approach consists of SimGNN [6], GOTSim [8], GraphSim [7], GEN [5] and GMN [5], which compare the node embeddings and find suitable alignments between them. Here, GMN applies cross attention based mechanism on the node embeddings given a graph neural network [49]. Recently, Chen et al. [57] designed a structure aware transformer architecture, which can represent a subgraph around a node more effectively than several other representation models.

**Differentiable solvers for combinatorial algorithms.** Our work attempts to find a neural surrogate for the combinatorial challenge of maximizing objective scores over a permutation space. In effect, we are attempting to solve an Optimal Transport problem, where the central challenge is to present a neural gadget which is differentiable and backpropagable, thus enabling end-to-end training. Cuturi [58] utilized iterative row and column normalization, earlier proposed by Sinkhorn [59], to approximately solve the transportation problem subject to marginal constraints. In another approach, Vlastelica et al. [60] attempted to solve combinatorial problems *exactly* using available black-box solvers, by proposing to use the derivative of an affine surrogate of the piece-wise constant function in the backward pass. Rolínek et al. [61] leverage this to perform deep graph matching based on explicit supervision of the ground truth node alignments. In another approach, Berthet et al. [62] perturb the inputs to the discrete solvers with random noise, so as to make them differentiable. Karalias and Loukas [63] design probabilistic loss functions for tackling the combinatorial objective of selecting a subset of nodes adhering to some given property. Finally, Kotary et al. [64] present a detailed survey of the existing neural approaches for solving constrained optimization problems on graphs.

## 2 Late interaction models for MCES and MCCS

In this section, we first write down the exact objectives for MCES and MCCS. These expressions are partly based upon a pairing of nodes between query and corpus graphs, and partly on typical graph algorithms. They lead naturally to our subsequent development of two late interaction models, LMCES and LMCCS. We begin with formal definitions of MCES and MCCS.

**Definition 1 (MCES and MCCS)** *Given query and corpus graphs $G_q=(V_q, E_q)$ and $G_c=(V_c, E_c)$.*

*(1) The maximum common edge subgraph $\mathrm{MCES}(G_q, G_c)$ is the common (not necessarily induced nor connected) subgraph between $G_q$ and $G_c$, having the maximum number of edges [15].*
*(2) The maximum common connected subgraph $\mathrm{MCCS}(G_q, G_c)$ is the common connected subgraph with the maximum number of nodes [21].*

### 2.1 Combinatorial formulations for MCES and MCCS

As mentioned in Section 1.2, combinatorial algorithms for MCES and MCCS abound in the literature [34, 33, 15, 18, 17, 30]. However, it is difficult to design neural surrogates for these algorithms. Therefore, we come up with the new optimization objectives for MCES and MCCS, which allow us to design neural surrogates by gradually replacing its different components with differentiable parameterized components.

**Exact MCES.** Given a query graph $G_q = (V_q, E_q)$ and a corpus graph $G_c = (V_c, E_c)$, we pad the graph having fewer vertices (typically, $G_q$), with $||V_c| - |V_q||$ disconnected nodes. This ensures that the padded graphs have an equal number of nodes $N$. Let us denote the adjacency matrices of $G_q$ and $G_c$ (after padding) as $\boldsymbol{A}_q \in \{0,1\}^{N \times N}$ and $\boldsymbol{A}_c \in \{0,1\}^{N \times N}$. To find $\mathrm{MCES}(G_q, G_c)$ from the

adjacency matrices $\boldsymbol{A}_q$ and $\boldsymbol{A}_c$, we first obtain the candidate common subgraph under some proposed node alignment given by permutations $\boldsymbol{P}$ of $\boldsymbol{A}_c$, which can be characterized by the adjacency matrix $\min(\boldsymbol{A}_q, \boldsymbol{P}\boldsymbol{A}_c\boldsymbol{P}^T)$. This matrix shows the overlapping edges under the proposed node alignment. Subsequently, we choose the permutation which maximizes the total number of edges in this subgraph. Formally, we compute the MCES score by solving a coverage maximization problem, as follows:

$$\max_{\boldsymbol{P} \in \mathcal{P}} \ \sum_{i,j} \min \left(\boldsymbol{A}_q, \boldsymbol{P}\boldsymbol{A}_c\boldsymbol{P}^\top\right)_{i,j} \tag{1}$$

where $\mathcal{P}$ is the set of permutation matrices of size $N \times N$ and the $\min$ operator is applied elementwise.

**Exact MCCS.** MCES does not require the common subgraph to be connected, which may be desirable in some applications. For example, in keyword search and question answering over knowledge graphs (KGs) [22, 65, 23], one may wish to have the entity nodes, forming the response, to be connected to each other. In molecule search, one may require a connected functional group to be present in both query and corpus graphs [19]. In such situations, MCCS may be more appropriate.

Given a query graph $G_q$ and a corpus graph $G_c$ and their adjacency matrices $\boldsymbol{A}_q$ and $\boldsymbol{A}_c$ after padding, we first apply a row-column permutation on $\boldsymbol{A}_c$ using the permutation matrix $\boldsymbol{P}$ and obtain the candidate common subgraph with adjacency matrix $\min(\boldsymbol{A}_q, \boldsymbol{P}\boldsymbol{A}_c\boldsymbol{P}^\top)$. Then we apply Tarjan's algorithm [66] to return the size of the largest connected component, which we maximize w.r.t. $\boldsymbol{P}$:

$$\max_{\boldsymbol{P} \in \mathcal{P}} \ \text{TARJANSCC}\left(\min\left(\boldsymbol{A}_q, \boldsymbol{P}\boldsymbol{A}_c\boldsymbol{P}^\top\right)\right). \tag{2}$$

Here TARJANSCC takes the adjacency matrix as input and returns the size of the largest connected component of corresponding graph.

**Bottleneck of approximating the optimization problems** (1) **and** (2)**.** One way of avoiding the intractability of searching through $\boldsymbol{P}$, is to replace the hard permutation matrix with a differentiable soft surrogate via the Gumbel-Sinkhorn (GS) network [67, also see Section B]. However, such a relaxation is not adequate on its own.

1. Most elements of $\min(\boldsymbol{A}_q, \boldsymbol{P}\boldsymbol{A}_c\boldsymbol{P}^\top)$ are binary, which deprives the learner of gradient signals.
2. In practice, the nodes or edges may have associated (noisy) features, which play an important role in determining similarity between nodes or edges across query and corpus graphs. For example, in scene graph based image retrieval, "panther" may be deemed similar to "leopard". The objectives in Eqs. (1) and (2) do not capture such phenomenon.
3. Tarjan's algorithm first applies DFS on a graph to find the connected components and then computes the size of the largest among them, in terms of the number of nodes. Therefore, even for a fixed $\boldsymbol{P}$, it is not differentiable.

Next, we address the above bottlenecks by replacing the objectives (1) and (2) with two neural surrogates, which are summarized in Figure 1.

## 2.2 Design of LMCES

We design the neural surrogate of Eq. (1) by replacing the adjacency matrices with the corresponding continuous node embeddings computed by a GNN, and the hard permutation matrix with a soft surrogate—a doubly stochastic matrix—generated by the Gumbel-Sikhorn network [67]. These node embeddings allow us to compute non-zero gradients and approximate similarity between nodes and their local neighborhood in the continuous domain. Specifically, we compute this neural surrogate in the following two steps.

**Step 1: Computing node embeddings.** We use a message passing graph neural network [49, 68] $\text{GNN}_\theta$ with $R$ propagation layers and trainable parameters $\theta$, to compute the node embeddings $\boldsymbol{h}_u(1), \ldots, \boldsymbol{h}_u(R) \in \mathbb{R}^d$, for each node $u$ in the query and corpus graphs, to which $\text{GNN}_\theta$ is applied separately. Finally, we build two matrices $\boldsymbol{H}_q(r), \boldsymbol{H}_c(r) \in \mathbb{R}^{N \times d}$ for $r \in [R]$ by stacking the node embedding vectors for query and corpus graphs. Formally, we have

$$\boldsymbol{H}_q(1), \ldots, \boldsymbol{H}_q(R) = \text{GNN}_\theta(G_q), \qquad \boldsymbol{H}_c(1), \ldots, \boldsymbol{H}_c(R) = \text{GNN}_\theta(G_c) \tag{3}$$

**Step 2: Interaction between $\boldsymbol{H}_q(r)$ and $\boldsymbol{H}_c(r)$.** In principle, the embeddings $\boldsymbol{h}_u(1), \ldots, \boldsymbol{h}_u(R)$ of a node $u$ capture information about the neighborhood of $u$. Thus, we can view the set of embedding matrices $\{\boldsymbol{H}_q(r) \,|\, r \in [R]\}$ and $\{\boldsymbol{H}_c(r) \,|\, r \in [R]\}$ as a reasonable representation of the query and corpus graphs, respectively. To compute a smooth surrogate of the adjacency matrix of the common subgraph, i.e., $\min(\boldsymbol{A}_q, \boldsymbol{P}\boldsymbol{A}_c\boldsymbol{P}^\top)$, we seek to find the corresponding alignment between $\boldsymbol{H}_q(r)$ and $\boldsymbol{H}_c(r)$ using soft-permutation (doubly stochastic) matrices $\boldsymbol{P}^{(r)}$ generated through a Gumbel-Sikhorn

network $\text{GS}_\phi$. Here, we feed $\boldsymbol{H}_q(r)$ and $\boldsymbol{H}_c(r)$ into $\text{GS}_\phi$ and obtain a doubly stochastic matrix $\boldsymbol{P}^{(r)}$:

$$\boldsymbol{P}^{(r)} = \text{GS}_\phi(\boldsymbol{H}_q(r), \boldsymbol{H}_c(r)) \quad \forall\, r \in [R] \tag{4}$$

Finally, we replace the true relevance scoring function in Eq. (1) with the following smooth surrogate:

$$s(G_q, G_c) = \sum_{r \in [R]} w_r \sum_{i,j} \min(\boldsymbol{H}_q(r), \boldsymbol{P}^{(r)}\boldsymbol{H}_c(r))_{i,j} \tag{5}$$

Here $\{w_r \geq 0 : r \in [R]\}$ are trainable parameters, balancing the quality of signals over all message rounds $r$. Note that the $R$ message rounds execute on the query and corpus graphs separately. The interaction between corresponding rounds, happens at the very end.

## 2.3 Design of LMCCS

In case of MCES, our key modification was to replace the adjacency matrices with node embeddings and design a differentiable network to generate a soft-permutation matrix. In the case of MCCS, we have to also replace the non-differentiable step of finding the size of the largest connected component of the common subgraph with a neural surrogate, for which we design a novel *gossip* protocol.

**Differentiable gossip protocol to compute the largest connected component.** Given any graph $G = (V, E)$ with the adjacency matrix $\boldsymbol{B}$, we can find its largest connected component (LCC) by using a gossip protocol. At iteration $t = 0$, we start with assigning each node a *message* vector $\boldsymbol{x}_u(0) \in \mathbb{R}^N$, which is the one-hot encoding of the node $u$, *i.e.*, $\boldsymbol{x}_u(0)[v] = 1$ for $v = u$ and 0 otherwise. In iteration $t > 0$, we update the message vectors $\boldsymbol{x}_u(t+1)$ as $\boldsymbol{x}_u(t+1) = \sum_{v \in \text{nbr}(u) \cup \{u\}} \boldsymbol{x}_v(t)$. Here, $\text{nbr}(u)$ is the set of neighbors of $u$. Initially we start with sparse vector $\boldsymbol{x}_u$ with exactly one non-zero entry. As we increase the number of iterations, $u$ would gradually collect messages from the distant nodes which are reachable from $u$. This would increase the number of non-zero entries of $\boldsymbol{x}_u$. For sufficiently large value of iterations $T$ (diameter of $G$), we will attain $\boldsymbol{x}_u(T)[v] \neq 0$ whenever $u$ and $v$ lie in the same connected component and $\boldsymbol{x}_u(T)[v] = 0$, otherwise. Once this stage is reached, one can easily compute the number of nodes in the largest connected component of $G$ as $\max_u ||\boldsymbol{x}_u(T)||_0$, *i.e.*,

---

**Algorithm 1** GOSSIP($\boldsymbol{B}$)

---
1: $\mathbf{X}(0) = \mathbb{I}$    # identity
2: **for** $t = 1, \ldots T - 1$ **do**
3:     $\mathbf{X}(t+1) \leftarrow \mathbf{X}(t)(\boldsymbol{B} + \mathbb{I})$
4: **Return** $\max_{u \in V} ||\mathbf{X}(T)[\bullet, u]||_0$

---

the maximum number of non-zero entries in a message vector. Algorithm 1 shows the gossip protocol, with the adjacency matrix $\boldsymbol{B}$ as input and the size of the largest connected component as output.

**Exact MCCS computation using the gossip protocol.** Given the query and corpus graphs $G_q$ and $G_c$ with their adjacency matrices $\boldsymbol{A}_q$ and $\boldsymbol{A}_c$, we can rewrite Eqn. (2) in the equivalent form

$$\max_{\boldsymbol{P} \in \mathcal{P}} \text{GOSSIP}(\min(\boldsymbol{A}_q, \boldsymbol{P}\boldsymbol{A}_c\boldsymbol{P}^\top)). \tag{6}$$

Recall that $\mathcal{P}$ is the set permutation matrices of size $N \times N$.

**Neural surrogate.** One can use a GS network to obtain a permutation matrix $\boldsymbol{P}$, which in principle, can support backpropagation of $\min(\boldsymbol{A}_q, \boldsymbol{P}\boldsymbol{A}_c\boldsymbol{P}^\top)$. However, as mentioned in Section 2.1 (items 1 and 2), $\boldsymbol{A}_\bullet$ is 0/1, and $\boldsymbol{P}$ often saturates, losing gradient signal for training. In response, we will design a neural surrogate for the true MCCS size given in Eq. (2), using three steps.

**Step 1: Computation of edge embeddings.** To tackle the above challenge, we introduce a parallel back-propagation path, in order to allow the GNNs to learn which edges are more important than others. To this end, we first use the GNNs to compute edge embedding vectors $\boldsymbol{m}_e(r) \in \mathbb{R}^{d_E}$ for edges $e \in E$, in addition to node embeddings at each propagation layer $r \in [R]$. Then, we gather the edge embeddings from the final layer $R$, into the matrices $\boldsymbol{M}_q(R), \boldsymbol{M}_c(R) \in \mathbb{R}^{|E| \times d_E}$ for query and corpus pairs. Next, we feed each separately into a neural network $L_\alpha$ with trainable parameters $\alpha$, which predicts the importance of edges based on each edge embedding. Thus, we obtain two matrices $\boldsymbol{S}_q \in \mathbb{R}^{N \times N}$ and $\boldsymbol{S}_c \in \mathbb{R}^{N \times N}$, which are composed of the edge scores between the corresponding node pairs. Formally, we have:

$$\boldsymbol{H}_q(R), \boldsymbol{M}_q(R) = \text{GNN}_\theta(G_q), \quad \boldsymbol{H}_c(R), \boldsymbol{M}_c(R) = \text{GNN}_\theta(G_c) \tag{7}$$

$$\boldsymbol{S}_q = L_\alpha\left(\boldsymbol{M}_q(R)\right), \quad \boldsymbol{S}_c = L_\alpha\left(\boldsymbol{M}_c(R)\right) \tag{8}$$

In our implementation, $L_\alpha$ consists of one linear layer, a ReLU layer and another linear layer.

**Step 2: Continuous approximation of MCCS.** In MCES, we approximated the MCES score in Eq. (1) directly, using the neural coverage function defined in Eq. (5) in terms of the node embeddings. Note that, here, we did not attempt to develop any continuous approximation of $\min\left(\boldsymbol{A}_q, \boldsymbol{P}\boldsymbol{A}_c\boldsymbol{P}^\top\right)$— the adjacency matrix of the candidate common subgraph. However, in order to apply our proposed

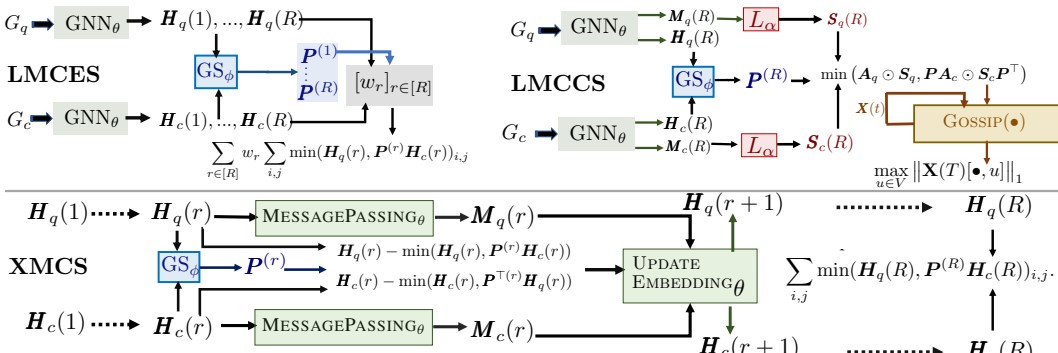

Figure 1: Proposed late (top row) and early (bottom row) interaction models. **Top Left**: LM-CES encodes the given query-corpus graphs separately, using $R$ layers of $\text{GNN}_\theta$ to compute node embedding matrices $\boldsymbol{H}_q(r), \boldsymbol{H}_c(r)$ with $r = \{1, \ldots, R\}$. For each layer $r$, the Gumbel-Sinkhorn network $\text{GS}_\phi$ computes the optimal alignment $\boldsymbol{P}^{(r)}$ between $\boldsymbol{H}_q(r)$ and $\boldsymbol{H}_c(r)$, which is finally used compute the relevance score in Eq. (5). **Top Right**: LMCCS uses $\text{GNN}_\theta$ to encode the $R$-hop node and edge embeddings, $\boldsymbol{H}_\bullet(R)$ and $\boldsymbol{M}_\bullet(R)$. The edge embeddings $\boldsymbol{M}_\bullet(R)$ are fed into $L_\alpha$ to predict edge score matrix $\boldsymbol{S}_\bullet(R)$ whose valid edge entries are encoded in $\boldsymbol{A} \odot \boldsymbol{S}$. As before, $\text{GS}_\phi(\boldsymbol{H}_q(R), \boldsymbol{H}_c(R))$ generates alignment $\boldsymbol{P}^{(R)}$ which is used to compute candidate MCS graph $\min(\boldsymbol{A}_q \odot \boldsymbol{S}_q, \boldsymbol{P}(\boldsymbol{A}_c \odot \boldsymbol{S}_c)\boldsymbol{P}^\top)$, which is then fed into the neural GOSSIP module, to predict the final MCCS score. **Bottom**: At each layer $r$, XMCS uses the alignment matrix $\boldsymbol{P}^{(r)}$ to factor the cross graph influences using $\boldsymbol{H}_q(r) - \min(\boldsymbol{H}_q(r), \boldsymbol{P}^{(r)}\boldsymbol{H}_c(r))$ for query, and $\boldsymbol{H}_c(r) - \min(\boldsymbol{H}_c(r), \boldsymbol{P}^{(r)^T}\boldsymbol{H}_q(r))$ for corpus. This is used to update the node embeddings $\boldsymbol{H}_\bullet(r) \to \boldsymbol{H}_\bullet(r+1)$. Embeddings from the final layer are used to compute the final relevance score.

gossip protocol (or its neural approximation), we need an estimate of the adjacency matrix of the common subgraph. Therefore, we compute the noisy estimator as follows:

$$\boldsymbol{P} = \text{GS}_\phi(\boldsymbol{H}_q(R), \boldsymbol{H}_c(R)), \qquad \boldsymbol{B} = \min\left(\boldsymbol{A}_q \odot \boldsymbol{S}_q, \boldsymbol{P}(\boldsymbol{A}_c \odot \boldsymbol{S}_c)\boldsymbol{P}^\top\right) \qquad (9)$$

In the above expression, we replace all the non-zero entries of $\boldsymbol{A}_q$ and $\boldsymbol{A}_c$ with the corresponding edge importance scores of $\boldsymbol{S}_q$ and $\boldsymbol{S}_c$. This facilitates backpropagation more effectively, as opposed to the original 0/1 adjacency representation of the common subgraph (item 1 in Section 2.1). Here, we generate the permutation matrix $\boldsymbol{P}$ using Gumbel-Sinkhorn network similar to MCES, except that here we generate only one permutation matrix based on the embeddings on the final layer, whereas in MCES, we generated permutations $\boldsymbol{P}^{(r)}$ for each layer $r$.

**Step 3: Neural implementation of gossip.** Our gossip protocol in Algorithm 1 is already differentiable. However, by the virtue of the way it is designed, it would give accurate results only if the input matrix consists of 0/1 values. However, our neural estimate $\boldsymbol{B}$ contains mostly non-zero continuous values and many of them can go beyond $\pm 1$. As a result, the resultant matrix $\mathbf{X}(t) = \mathbf{X}(0)(\boldsymbol{B} + \mathbb{I})^t$ in Algorithm 1 may suffer from extremely poor conditioning. To tackle this problem, we use a *noise filter* network at the final step $T$. We first estimate a dynamic threshold $\tau \in [0, \infty)$ and then set the values of $\mathbf{X}(T)$ below that threshold to zero. Finally, we scale the non-zero entries between $(0, 1)$ using a sigmoid activation $\sigma(\cdot)$. Formally, we define:

$$\tau = \text{THRESH}_\beta(\mathbf{X}(T)); \qquad \hat{\mathbf{X}}(T) = 2\sigma\left(\text{ReLU}[\mathbf{X}(T) - \tau]/\lambda\right) - 1, \qquad (10)$$

where $\lambda$ is a temperature hyperparameter. $\text{THRESH}_\beta$ consists of a linear, a ReLU, then another linear layer. Note that $\hat{\mathbf{X}}(T) \in [0, 1]$ by design, which lets us replace the $L_0$ norm in the final score (Algorithm 1, line 4) with the more benign $L_1$ norm, followed by a maxpool over nodes:

$$s(G_q, G_c) = \max_{u \in V} \left\|\hat{\mathbf{X}}(T)[\bullet, u]\right\|_1 \qquad (11)$$

Note that the interaction steps 2 and 3 above are completely decoupled from step 1 and do not have any contribution in computing embeddings.

## 3 Early/cross interaction model: XMCS

Although late interaction models offer efficient training and fast inference, prior work [5] suggests that early interaction models, while slower, may offer better accuracy. Motivated by such successes,

we propose a unified early interaction model, called XMCS, which works for both MCES and MCCS. XMCS is slower than LMCES and LMCCS, but provides significant accuracy gains (Section 4). Moreover, it is significantly faster than GMN [5]. As before, the relevance score $s(G_q, G_c)$ is computed using a graph neural network (GNN) with $R$ propagation layers, but each graph influences embeddings of the other graph in each layer.

**Initialization of node embeddings.** We start with the raw node features $\boldsymbol{z}_u$ for each node $u \in V_q \cup V_c$ and use them to initialize the node embeddings $\boldsymbol{h}_u$.

$$\boldsymbol{h}_u(0) = \text{FEATUREENCODER}_\theta(\boldsymbol{z}_u) \tag{12}$$

**Embedding computation via interaction between $G_q$ and $G_c$.** Given the node embeddings $\boldsymbol{h}_u(r)$ for some propagation layer $r < R$, we first encode the intra-graph influences across all edges in both query and corpus graphs. Accordingly, we obtain directed message vectors, for each pair of nodes $(u, v) \in E_q \cup E_c$, which are then aggregated using a simple sum aggregator.

$$\boldsymbol{m}_{uv}(r) = \text{MESSAGEPASSING}_\theta(\boldsymbol{h}_u(r), \boldsymbol{h}_v(r)) \quad \forall (u, v) \in E_q \cup E_c \tag{13}$$

$$\overline{\boldsymbol{m}}_u(r) = \sum_{v \in \text{nbr}(u)} \boldsymbol{m}_{uv}(r) \quad \forall u \in V_q \cup V_c \tag{14}$$

Next, we perform the interaction step across the query and corpus graphs $G_q$ and $G_c$, using a graph alignment network, which is modeled using $\text{GS}_\phi$, similar to the late interaction models in Eq. (4). We build embedding matrices $\boldsymbol{H}_q(r)$ and $\boldsymbol{H}_c(r)$ by stacking $\boldsymbol{h}_u(r)$ from $G_q$ and $G_c$ respectively. Then, we feed them into $\text{GS}_\phi$ to generate an alignment matrix, and finally compute the difference of the query and the corpus graphs from the underlying MCS in the continuous embedding space:

$$\boldsymbol{P}^{(r)} = \text{GS}_\phi(\boldsymbol{H}_q(r), \boldsymbol{H}_c(r)); \qquad \boldsymbol{\Delta}_q(r) = \boldsymbol{H}_q(r) - \min(\boldsymbol{H}_q(r), \boldsymbol{P}^{(r)}\boldsymbol{H}_c(r));$$

$$\boldsymbol{\Delta}_c(r) = \boldsymbol{H}_c(r) - \min(\boldsymbol{H}_c(r), \boldsymbol{P}^{(r)\top}\boldsymbol{H}_q(r)). \tag{15}$$

Note that $\boldsymbol{\Delta}_q(r)$ in the above can also be written as $\text{ReLU}[\boldsymbol{H}_q(r) - \boldsymbol{P}^{(r)}\boldsymbol{H}_c(r)]$ (because $\min(a, b) = a - \text{ReLU}(a - b)$) and thus $\boldsymbol{\Delta}_q(r)$ (similarly, $\boldsymbol{\Delta}_c(r)$) captures the representation of a subgraph present in $G_q$ ($G_c$), which is not present in $G_c$ ($G_q$). Here, $\boldsymbol{P}$ provides an injective mapping from $V_c$ to $V_q$, in contrast to attention-based GMN [5], which is non-injective—it assigns one corpus node to one query node but one query node to possibly multiple corpus nodes.

Next, the node embeddings $\boldsymbol{h}_u$ are updated using aggregated intra-graph and cross-graph influences.

$$\boldsymbol{h}_u(r+1) = \text{UPDATEEMBEDDING}_\theta\big(\boldsymbol{h}_u(r), \overline{\boldsymbol{m}}_u(r), \sum_j \boldsymbol{\Delta}(r)[u, j]\big) \quad \forall u \in V_q \cup V_c \tag{16}$$

The node embeddings of $G_q$ explicitly depend on $G_c$ via the alignment $\boldsymbol{P}^{(\bullet)}$ and vice-versa.

**Relevance score computation.** Finally, we compute the relevance score using the neural surrogate:

$$s(G_q, G_c) = \sum_{i,j} \min(\boldsymbol{H}_q(R), \boldsymbol{P}^{(R)}\boldsymbol{H}_c(R))_{i,j}. \tag{17}$$

Clearly, the above scoring function directly approximates the MCES objective (1) similar to the score given by LMCES in Eq. (5), except that here we use the embeddings at the last layer to compute the score. Although one can subsequently a combine gossip network with the above model, we found that it does not improve accuracy significantly and moreover, results in extreme slowdown.

## 4 Experiments

In this section, we provide a comprehensive evaluation of our models across seven datasets and show that they outperform several competitors [6, 7, 8, 10, 11, 5]. Our code is in https://tinyurl.com/mccs-xmcs.

### 4.1 Experimental setup

**Datasets.** We experiment with seven datasets, *viz.*, MSRC-21 (MSRC), PTC-MM (MM), PTC-FR (FR), PTC-MR (MR), PTC-FM (FM), COX2 (COX) and DD. The details about them are described in Appendix C. Among these datasets, we report the results of the first six datasets in the main paper and the DD dataset in Appendix D. For every dataset, we have a corpus graph database with 800 graphs and a set of 500 query graphs, leading to 400000 query-corpus pair of graphs.

**State-of-the-art methods compared.** We compare our method against six state-of-the-art late interaction models, *viz.*, (i) SimGNN [6], (ii) GraphSim [7], (iii) GOTSim [8], (iv) NeuroMatch [10], (v) IsoNet [11] and (vi) Graph embedding network (GEN) [5]; and one early interaction model, *viz.*,

| MCES | MSE (lower is better) | | | | | | KTau (higher is better) | | | | | |
|---|---|---|---|---|---|---|---|---|---|---|---|---|
| | MSRC | MM | FR | MR | FM | COX | MSRC | MM | FR | MR | FM | COX |
| **Late** SimGNN | 0.910 | 0.302 | 0.355 | 0.337 | 0.331 | 0.281 | 0.232 | 0.368 | 0.358 | 0.354 | 0.372 | 0.394 |
| GraphSim | 0.629 | 0.274 | 0.282 | 0.274 | 0.261 | 0.249 | 0.461 | 0.432 | 0.458 | 0.454 | 0.500 | 0.403 |
| GOTSim | 0.496 | 0.343 | 0.326 | 0.320 | 0.359 | 0.328 | 0.564 | 0.464 | 0.448 | 0.516 | 0.496 | 0.374 |
| NeuroMatch | 0.582 | 0.308 | 0.282 | 0.795 | 0.604 | 0.269 | 0.632 | 0.488 | 0.516 | 0.548 | 0.535 | 0.514 |
| IsoNet | 0.276 | 0.225 | 0.220 | 0.209 | 0.253 | 0.182 | 0.669 | 0.506 | 0.504 | 0.537 | 0.532 | 0.522 |
| GEN | 0.426 | 0.311 | 0.273 | 0.284 | 0.324 | 0.277 | 0.627 | 0.416 | 0.468 | 0.456 | 0.456 | 0.466 |
| LMCES | 0.232 | 0.167 | 0.170 | 0.162 | 0.163 | 0.140 | 0.691 | 0.577 | 0.588 | 0.598 | 0.610 | 0.574 |
| **Early** GMN | 0.269 | 0.184 | 0.181 | 0.178 | 0.189 | 0.155 | 0.670 | 0.544 | 0.567 | 0.568 | 0.569 | 0.555 |
| XMCS | 0.226 | 0.154 | 0.162 | 0.154 | 0.160 | 0.132 | 0.699 | 0.582 | 0.594 | 0.612 | 0.606 | 0.580 |

| MCCS | MSE (lower is better) | | | | | | KTau (higher is better) | | | | | |
|---|---|---|---|---|---|---|---|---|---|---|---|---|
| | MSRC | MM | FR | MR | FM | COX | MSRC | MM | FR | MR | FM | COX |
| **Late** SimGNN | 0.100 | 0.360 | 0.337 | 0.233 | 0.316 | 0.289 | 0.125 | 0.281 | 0.308 | 0.313 | 0.299 | 0.366 |
| GraphSim | 0.088 | 0.283 | 0.290 | 0.221 | 0.255 | 0.325 | 0.153 | 0.336 | 0.337 | 0.315 | 0.366 | 0.292 |
| GOTSim | 0.165 | 0.416 | 0.340 | 0.330 | 0.321 | 0.318 | -0.088 | 0.320 | 0.327 | 0.307 | 0.380 | 0.416 |
| NeuroMatch | 0.352 | 0.376 | 0.326 | 0.351 | 0.295 | 0.984 | 0.125 | 0.376 | 0.365 | 0.370 | 0.406 | 0.440 |
| IsoNet | 0.086 | 0.237 | 0.244 | 0.191 | 0.218 | 0.253 | 0.185 | 0.381 | 0.388 | 0.351 | 0.402 | 0.406 |
| GEN | 0.171 | 0.366 | 0.344 | 0.290 | 0.356 | 0.309 | 0.111 | 0.325 | 0.332 | 0.305 | 0.326 | 0.391 |
| LMCCS | 0.068 | 0.174 | 0.179 | 0.134 | 0.173 | 0.177 | 0.248 | 0.451 | 0.438 | 0.406 | 0.457 | 0.487 |
| **Early** GMN | 0.101 | 0.200 | 0.216 | 0.156 | 0.193 | 0.176 | 0.174 | 0.416 | 0.405 | 0.379 | 0.431 | 0.479 |
| XMCS | 0.071 | 0.168 | 0.163 | 0.131 | 0.168 | 0.153 | 0.198 | 0.452 | 0.451 | 0.412 | 0.453 | 0.501 |

Table 1: Performance measured using mean square error (MSE) (left half) and Kendall-Tau Rank Correlation (Ktau) (right half) of our models and state-of-the-art baselines, *viz.*, SimGNN [6], GraphSim [7], GOTSim [8], Neuromatch [10], IsoNet [11], GEN [5], GMN [5] on 20% test set. Top-half and bottom-half report results for MCES and MCCS respectively. Except GMN, all SOTA methods are late interaction models. Numbers in green (blue) indicate the best performers among early (late) interaction models. Numbers in yellow indicate second best performers for late interaction models.

(vii) Graph Matching Network (GMN) [5]. All methods use a general purpose scoring layer, except for NeuroMatch and IsoNet, which are specifcially designed for subgraph isomorphism.

**Training and evaluation.** Given corpus graphs $C = \{G_c\}$, query graphs $Q = \{G_q\}$ and their gold MCES and MCCS values $\{y_{\text{MCES}}(G_q, G_c)\}$ and $\{y_{\text{MCCS}}(G_q, G_c)\}$, for $G_q \in Q, G_c \in C$, we partition the query set into 60% training, 20% validation and 20% test folds. We train all methods by minimizing mean square error (MSE) loss between the predicted output and gold MCS (MCES or MCCS) values on the training split.

$$\min_\Lambda \sum_{G_q \in Q, G_c \in C} (s_\Lambda(G_q, G_c) - y(G_q, G_c))^2 \tag{18}$$

Here $y$ can be either $y_{\text{MCES}}$ or $y_{\text{MCCS}}$ and $\Lambda$ is the set of trainable parameters for the relevance scoring function $s_\Lambda(G_q, G_c)$. In the context of our models, $\Lambda = \{\theta, \phi, w_{1:R}\}$ for LMCES, $\Lambda = \{\theta, \phi, \alpha, \beta\}$ for LMCCS and $\Lambda = \{\theta, \phi\}$ for XMCS model. We use the validation split to tune various hyperparameters. Subsequently, we use the trained models to predict MCS scores between the test query graphs and the corpus graphs. For each of the query graphs in the test split, we use the predicted outputs to compute the MSE and Kendall-Tau rank correlation (KTau) values. Finally, we report the average MSE and mean KTau values across all the test query graphs.

## 4.2 Results

**Comparison with state-of-the-art methods.** In Table 1, we compare the performance of LMCES and XMCS (LMCCS and XMCS) against the state-of-the-art methods on both MCES (MCCS) based retrieval tasks. We observe: **(1)** For MCES and MCCS tasks, LMCES and LMCCS outperform all late interaction models by a substantial margin in terms of MSE and KTau across all datasets. **(2)** XMCS consistently outperforms GMN, the only early interaction baseline model. Suprisingly, GMN is also outperformed by our late interaction models, except in COX for the MCCS task. The likely reason is that GMN uses a general purpose scoring function and a cross attention mechanism that induces a non-injective mapping between the nodes and edges of the query corpus pairs. **(3)** XMCS outperforms both LMCES and LMCCS, as expected. **(4)** For both MCES and MCCS tasks, IsoNet is consistently second-best with respect to MSE. IsoNet is a subgraph isomorphism based retrieval model and its scoring function is proportional to $\sum_{i,j} \text{ReLU}(\boldsymbol{M}_q(R) - \boldsymbol{P}^{(R)} \boldsymbol{M}_c(R))_{i,j}$, which can be re-written as $\sum_{i,j} [\boldsymbol{M}_q(R) - \min(\boldsymbol{M}_q(R), \boldsymbol{P}^{(R)} \boldsymbol{M}_c(R))_{i,j}]$ (since $\min(a, b) = a - \text{ReLU}(a - b)$). Thus, the second term captures MCES score. During training, IsoNet is also able to shift and scale the above score to offset the additional term involving $\boldsymbol{M}_q(R)$, which likely allows it to outperform other

| | **MCES** | MSRC | MM | FR | MR |
|---|---|---|---|---|---|
| **Late** | GEN | 0.426 | 0.311 | 0.273 | 0.284 |
| | GEN (MCS) | 0.284 | 0.181 | 0.179 | 0.169 |
| | IsoNet | 0.276 | 0.225 | 0.220 | 0.209 |
| | IsoNet (MCS) | 0.260 | 0.187 | 0.178 | 0.173 |
| | LMCES | 0.232 | 0.167 | 0.170 | 0.162 |
| **Early** | GMN | 0.269 | 0.184 | 0.181 | 0.178 |
| | GMN (MCS) | 0.228 | 0.155 | 0.158 | 0.157 |
| | XMCS | 0.226 | 0.154 | 0.162 | 0.154 |

| | **MCCS** | MSRC | MM | FR | MR |
|---|---|---|---|---|---|
| **Late** | GEN | 0.171 | 0.366 | 0.344 | 0.290 |
| | GEN (MCS) | 0.076 | 0.226 | 0.195 | 0.161 |
| | IsoNet | 0.086 | 0.237 | 0.244 | 0.191 |
| | IsoNet (MCS) | 0.088 | 0.230 | 0.225 | 0.161 |
| | LMCCS | 0.068 | 0.174 | 0.179 | 0.134 |
| **Early** | GMN | 0.101 | 0.200 | 0.216 | 0.156 |
| | GEN (MCS) | 0.070 | 0.178 | 0.173 | 0.125 |
| | XMCS | 0.071 | 0.168 | 0.163 | 0.131 |

Table 2: Effect of replacing the general-purpose scoring layers with new layers customized to MCS on most competitive baselines, *viz.*, GEN and IsoNet (late interaction models) and GMN, across first four datasets (MSE). Numbers in green (red) indicate the best (second best) performers for early interaction models. Numbers in blue (yellow) indicate the best (second best) performers for late interaction models. The proposed modification improves performance of all baselines. However our models outperform them, even after modifying their layers, in most cases.

baselines. **(5)** Neuromatch is excellent with respect to KTau, where it is the second best performer in six out of twelve settings. This is due to NeuroMatch's order-embedding training objective, which translates well to the KTau rank correlation scores.

**Effect of replacing general purpose scoring layers with MCS customized layer.** The state-of-the-art methods use a general purpose scoring functions, whereas those of our models are tailored to MCS based retrieval. In order to probe the effect of such custom MCS scoring layers, we modify the three most competitive baselines, *viz.*, IsoNet, GEN, GMN, where we replace their scoring layers with a layer tailored for MCS. Specifically, for IsoNet, we set $s(G_q, G_c) = \sum_{i,j} \min(\boldsymbol{M}_q(R), \boldsymbol{P}^{(R)} \boldsymbol{M}_c(R))_{i,j}$ and for GEN and GMN, we use $s(G_q, G_c) = \sum_i \min(\boldsymbol{h}_q(R), \boldsymbol{h}_c(R))_i$. Table 2 summarizes the results, which show that: **(1)** All three baselines enjoy a significant accuracy boost from the MCS-tailored scoring layer. **(2)** LMCES and LMCCS continue to outperform MCS-tailored variants of late interaction baselines. XMCS outperforms MCS-tailored GMN in a majority of cases.

**Ablation study.** We consider four additional variants: (i) LMCES (final layer) where the relevance score is computed using only the embeddings of the $R^{\text{th}}$ layer, (ii) LMCCS (no gossip), where we remove the gossip network and compute $s(G_q, G_c) = \sum_{i,j} \min(\boldsymbol{A}_q \odot \boldsymbol{M}_q, \boldsymbol{P}^{(R)} \boldsymbol{A}_c \odot \boldsymbol{M}_c \boldsymbol{P}^{(R)})_{i,j}$, (iii) LMCCS (no NOISE FILTER) where we set $\tau_t = 0$ in Eq. (10) and (iv) XMCS (all layers) where we compute the relevance score in Eq. (17) using embeddings from all $R$ layers. Table 3 summarizes the results where the numbers in green (red) for early interaction

| | **MCES** | MSRC | MM | FR |
|---|---|---|---|---|
| **Late** | LMCES (final layer) | 0.237 | 0.175 | 0.170 |
| | LMCES | 0.232 | 0.167 | 0.170 |
| **Early** | XMCS (all layers) | 0.224 | 0.154 | 0.165 |
| | XMCS | 0.226 | 0.154 | 0.162 |
| | **MCCS** | MSRC | MM | FR |
| **Late** | LMCCS (no gossip) | 0.166 | 0.241 | 0.240 |
| | LMCCS (no NOISE FILTER) | 0.068 | 0.194 | 0.206 |
| | LMCCS | 0.068 | 0.174 | 0.179 |
| **Early** | XMCS (all layers) | 0.069 | 0.181 | 0.171 |
| | XMCS | 0.071 | 0.168 | 0.163 |

Table 3: Ablation study (MSE).

models, and blue (yellow) for late interaction models, indicate the best (second best) performers. We observe: **(1)** The scoring function of LMCES computed using Eq. (5) improves the performance for MSRC and MM datasets. **(2)** The gossip component is the most critical part of LMCCS. Upon removal of this component, the MSE of LMCCS significantly increases—for MSRC, it more than doubles. **(3)** The noise filter unit described in Eq. (10) is also an important component of LMCCS. Removal of this unit renders noticeable rise in MSE in MM and FR datasets. **(4)** Given an already high modeling power of XMCS, we do not observe any clear advantage if we use embeddings $\{\boldsymbol{H}_q(r), \boldsymbol{H}_c(r), r \in [R]\}$ from all $R$ layers to compute the final score in Eq. (17).

**Recovering latent linear combinations of MCES and MCCS scores.** In some applications, the ground truth relevance scores may be unknown or equal to a noisy latent function MCCS and MCES size. To simulate this scenario, here, we evaluate the performance when the ground truth relevance label is a convex combination of MCCS and MCES scores, *i.e.*, $a \cdot \text{MCCS-size} + (1-a) \cdot \text{MCES-size}$. We experiment with our late (LMCES and LMCCS) and early (XMCS) interaction models, as well as the baseline late (GEN) and early(GMN) interaction models equipped with our custom MCS layer. Additionally, we implement a new model COMBO, whose relevance score $s(G_q, G_c) = \sum w_1 s_{\text{LMCCS}}(G_q, G_c) + w_2 s_{\text{LMCES}}(G_q, G_c)$. Here, $\{w_r \geq 0 : r \in [2]\}$ are trainable parameters, which attempt to balance the signals from the LMCCS and LMCES scoring functions, in order to

| Combined | MSE (lower is better) | | | | KTau (higher is better) | | | |
|---|---|---|---|---|---|---|---|---|
| | MSRC | | MM | | MSRC | | MM | |
| | $a = 0.3$ | $a = 0.7$ | $a = 0.3$ | $a = 0.7$ | $a = 0.3$ | $a = 0.7$ | $a = 0.3$ | $a = 0.7$ |
| Late — GEN (MCS) | 0.150±0.004 | 0.071±0.008 | 0.146±0.005 | 0.165±0.012 | 0.664±0.004 | 0.644±0.004 | 0.567±0.005 | 0.549±0.005 |
| Late — LMCES | 0.125±0.004 | 0.066±0.008 | 0.143±0.006 | 0.177±0.013 | 0.692±0.004 | 0.675±0.004 | 0.587±0.005 | 0.565±0.005 |
| Late — LMCCS | 1.044±0.069 | 0.230±0.011 | 0.172±0.007 | 0.166±0.011 | 0.553±0.010 | 0.528±0.011 | 0.561±0.006 | 0.554±0.006 |
| Late — COMBO | 0.130±0.004 | 0.068±0.008 | 0.145±0.006 | 0.140±0.012 | 0.687±0.004 | 0.665±0.004 | 0.580±0.005 | 0.578±0.005 |
| Early — GMN (MCS) | 0.124±0.003 | 0.060±0.007 | 0.129±0.005 | 0.130±0.009 | 0.692±0.004 | 0.677±0.004 | 0.587±0.005 | 0.574±0.005 |
| Early — XMCS | 0.121±0.003 | 0.063±0.008 | 0.124±0.005 | 0.129±0.009 | 0.695±0.004 | 0.671±0.004 | 0.593±0.005 | 0.572±0.005 |

Table 4: Performance when **the ground truth is the convex combination of MCES and MCCS sizes**, *i.e.*, $a \cdot$ MCCS-size $+ (1-a) \cdot$ MCES-size, for two values of $a$ *viz.*, $a \in \{0.3, 0.7\}$. Performance measured using mean square error (MSE) (left half) and Kendall-Tau Rank Correlation (Ktau) (right half) with standard error, of our models and the baselines GEN and GMN, for two datasets. Numbers in green (red) indicate the best (second best) performers for early interaction models. Numbers in blue (yellow) indicate the best (second best) performers for late interaction models.

predict any given combination of ground truths. In Table 4, we report the performance in terms of MSE and KTau with standard error, for two latent (hidden from the learner) values of $a$, *viz.*, $a = 0.3$ and $a = 0.7$. We make the following observations:

1. GEN (MCS) is consistently outperformed in all cases, by one of our late interaction variants LMCES, LMCCS, or COMBO.
2. LMCCS is seen to be very susceptible to noise. It does not perform well even for $a = 0.7$ (30% MCES noise). In fact, its performance deteriorates rapidly for MCES dataset, due to the noisy MCES signals, with the performance for $a = 0.3$ being significantly worse than $a = 0.7$.
3. LMCES is seen to be more robust to noise, and is the best performer in six out of eight cases.
4. COMBO is overall seen to be the most well-balanced in terms of performance. Although it is the best performer in two out of eight cases, it is the second best close behind LMCES, for the remaining cases.
5. XMCS is seen to be able to adapt well to this noisy setup, and performs better than GMN (MCS) in five out of eight cases.

It is encouraging to see that XMCS does not need customization based on connectedness of the MCS, but we remind the reader that late interaction methods are substantially faster and may still have utility.

**Inference times.** Tables 1 and 2 suggest that GMN is the most competitive baseline. Here, we compare the inference time, on our entire test fold, taken by GMN against our late and early interaction models. Table 5 shows the results. We observe: **(1)** XMCS is 3× faster than GMN. This is because GMN's cross interaction demands processing one query-corpus pair at a time to account for variable graph sizes. In contrast, our Gumbel-Sinkhorn permutation network allows us to pad the adjacency matrices for

| LMCCS | LMCCS (no gossip) | LMCES | XMCS | GMN |
|---|---|---|---|---|
| 13.78 | 13.96 | 28.13 | 31.10 | 99.54 |

Table 5: Inference time (s), increasing from left to right.

batched processing, which results in significant speedup along with accuracy improvements. **(2)** LM-CES is 2× slower than LMCCS (no gossip), as it has to compute the permutation matrices for all $R$ layers. Furthermore, LMCCS and LMCCS (no gossip) require comparable inference times.

## 5 Conclusion

We proposed late (LMCES, LMCCS) and early (XMCS) interaction networks for scoring corpus graphs with respect to query graphs under a maximum common (connected) subgraph consideration. Our formulations depend on the relaxation of a node alignment matrix between the two graphs, and a neural 'gossip' protocol to measure the size of connected components. LMCES and LMCCS are superior with respect to both speed and accuracy among late interaction models. XMCS is comparable to the best early interaction models, while being much faster. Our work opens up interesting avenues for future research. It would be interesting to design neural MCS models which can also factor in similarity between attributes of nodes and edges. Another interesting direction is to design neural models for MCS detection across multiple graphs.

**Acknowledgement.** Indradyumna acknowledges PMRF fellowship and Qualcomm Innovation Fellowship. Soumen Chakrabarti and Abir De acknowledge IBM AI Horizon grant. Abir De also acknowledges DST Inspire grant.

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
