# OpenReview forum: "Maximum Common Subgraph Guided Graph Retrieval: Late and Early Interaction Networks"
_NeurIPS.cc/2022/Conference — NeurIPS 2022 Accept_

### Official Review · Reviewer_pKLh · 2022-07-09

**Rating:** 7
**Confidence:** 3
**Soundness:** 3 good
**Presentation:** 3 good
**Contribution:** 4 excellent

**Summary:**

This paper studies the neural graph retrieval problem. Specifically, it focuses on two similarilty scores, MCCS and MCES. Since exact computation is intractable, thus this paper designs neural functions to approximate MCES and MCCS. This paper proposes both early and late fusion of neural graph retriever, and experiments overs seven datasets show that the proposed approaches outperform baselines on both accuracy and speed.

**Questions:**

1. Does different GNN architecture affect the results?
2. Maybe add an experiment on hybrid interaction (start with late interaction, then interaction), and experiment on both accuracy and time. (see details in strengths/weaknesses).
3. Maybe add a few related works on neural information retrieval (e.g., dense retrieval)

**Limitations:**

Not addressed, but I do not think of any potential negative social impact.

**Strengths And Weaknesses:**

Strength:

1. Although several research work has focused on neural information retrieval, few works on graph neural retrieval, an important but relatively underexplored problem. This paper fills in the gap.
2. This paper proposed a simple neural graph retrieval method and works well on both accuracy and efficiency (significantly outperforms baselines)
3. The paper is well-written, it's enjoyable reading it.

Weakness (more like suggestions, not really a weak point)

Since this paper proposes both early and late interaction approaches, a natural idea is to combine both approaches (e.g., hybrid approach).
A straightforward approach is to start with late interaction to find some relevant graphs, then use early interaction to re-rank these graphs. The expectation is to find the optimal trade-off between accuracy and efficiency.

---

> ### Author Response · Authors · 2022-08-02
> **Response to Reviewer pKLh**
>
> > *  Does different GNN architecture affect the results?
>
> Our empirical analysis revealed that our graph encoder module provides injective mappings for the k-hop structural neighborhood around each node. Therefore, two node embeddings are equal only when the local subgraphs around the two nodes are structurally isomorphic. As such, injective GNNs can provide good accuracy in our problems. For example, Xu et al [1] proposed the Graph Information Network (GIN) which, under certain conditions,  is shown to be as powerful as the Weisfeiler & Lehman (1968) (WL) heuristic in distinguishing graph structures.
> We observe that when we use GIN architecture, we have MSE=0.181. On the other hand, we also worked with vanilla GCN (Kipf et al ICLR 2016) which provided MSE = 0.2050. Finally, our method provided MSE = 0.167. These results show that injective node mapping (our method, GIN) is more useful than non-injective node mapping (vanilla GCN) in this context.
>
> [1] Xu K, Hu W, Leskovec J, Jegelka S. How powerful are graph neural networks?.
>
>
> > *   Maybe add an experiment on hybrid interaction (start with late interaction, then interaction), and experiment on both accuracy and time. (see details in strengths/weaknesses).
>
> Following the reviewer’s suggestions, we have tried a hybrid 2-stage approach as follows. First, the late interaction model LMCCS is used to score all 800 corpus graphs for each query graph. Then for each query, the corpus graphs with top-50 scores are further re-scored using the early interaction model XCMS. We judge the effectiveness of this approach by using the MRR metric, which we feel is best suited to gauge correctness of ranking at the topmost positions. For each query, the corpus graphs with the highest MCS score is deemed relevant (binary label 1), and the remaining corpus graphs are deemed irrelevant (label 0). We present the result of our experiments on the MSRC dataset where the ground truth is a noisy latent combination of the MCCS and MCES sizes.
>
> | MSRC     | LMCCS | XMCS  | Hybrid |
> | -------- | ----- | ----- | ------ |
> | MRR      | 0.364 | 1.000 | 0.866  |
> | TIME (s) | 16.7  | 36.3  | 21.83  |
>
>
> We make the following observations:
> (1) The inference time of the late interaction model LMCCS is 2X faster than that of the early interaction model.
> (2) The MRR performance of LMCCS is significantly poorer than that of XMCS
> (3) The 2-stage inference process incurs an additional time of around 5 seconds, on top of the inference time of LMCCS. However, the total time of the 2 stage inference is still 1.7X faster than that of XMCS. Additionally, the MRR performance of the 2-stage inference is a significant improvement compared to that of LMCCS.
>
> > *  Maybe add a few related works on neural information retrieval (e.g., dense retrieval)
>
> Our transition from combinatorial to neural graph search methods mirrors a similar successful movement from venerated sparse text search methods (TFIDF, BM25) to dense neural text search methods (see citations below).
>
> Pretrained Transformers for Text Ranking: BERT and Beyond
> https://arxiv.org/abs/2010.06467
>
> An Introduction to Neural Information Retrieval
> https://www.microsoft.com/en-us/research/uploads/prod/2017/06/fntir2018-neuralir-mitra.pdf
>
> ColBERT: Efficient and Effective Passage Search via Contextualized Late Interaction over BERT
> https://cs.stanford.edu/~matei/papers/2020/sigir_colbert.pdf

---

### Official Review · Reviewer_UMf5 · 2022-07-09

**Rating:** 6
**Confidence:** 4
**Soundness:** 3 good
**Presentation:** 3 good
**Contribution:** 3 good

**Summary:**

The paper proposes new neural network architectures for the task of graph retrieval (computing a score between a query graph and a corpus graph) tailored for the Maximum Common Subgraph (MCS) metrics. Two late interaction models and an early interaction model are proposed with trade-off in efficiency and accuracy. Experiments on seven real-world datasets show the superiority of the proposed models.

**Questions:**

Please see above for questions.

**Limitations:**

The authors do not explicitly mention societal impact in details.

**Strengths And Weaknesses:**

Strengths:
1. The proposed method is relatively novel, albeit heavily relying on the Gumbel-Sinkhorn (GS) network.
2. Two variants in MCS definition are discussed (MCES and MCCS), which reflect the different needs of real-world applications.
3. The paper is well-written with enough details as well as code provided for reproducing the experimental results.

Weaknesses:
1. It is unclear how scalable the proposed methods are. First of all, the theoretical time and space complexity are not explicitly discussed, and thus the speedup compared with GMN is not explained from a theoretical point of well. Second, the statistics of the number of nodes and edges of the graphs in the seven datasets are not shown. Instead, the authors give a relatively vague description of how the graph pairs are generated in Section D, “Subsequently, we augment the seed query graph G′q , with randomly connected nodes and edges, which gives us the final query graph Gq for MCS computation.” How many nodes and edges are randomly added to the seed? What is the final size of the graph? This concern is of particular relevance given that exact MCS is NP-hard to compute, but the authors can generate the ground-truth MCS values “using the combinatorial formulations for exact MCCS and MCES,” raising the concern that whether the graph sizes are quite small so that even exact MCS computation is affordable. Figures 2 and 3 also show quite small graph pairs with ~20 nodes. Why would people use such approximate methods for MCS scores if the input graphs are small and the exact solvers can already handle them reasonably well? Without large graph experiments, the exact solver may be used as a baseline too, which is missing in the current experiments.
2. A minor concern is the applicability of the proposed models, since the models are specifically designed for MCS metrics, more specifically, MCCS and MCES. I agree that MCS metrics are already important and worth specific models designed for them, but it would be interesting to discuss the potential issues if the proposed methods are used for other metrics, e.g. GED, or domain-specific similarity scores. This would make the contribution and the scope of usage clearer.

---

> ### Author Response · Authors · 2022-08-02
> **Response to Reviewer UMf5  (Part 1)**
>
> > *It is unclear how scalable the proposed methods are. First of all, the theoretical time and space complexity are not explicitly discussed, and thus the speedup compared with GMN is not explained from a theoretical point well.
>
> During training, the number of epochs needed to optimize a non-convex objective cannot be analyzed or predicted in most applications. So we will focus on inference time, which is anyway more critical for search applications. Our late interaction methods LMCES and  LMCCS are more scalable than GMN because in every layer, when GMN computes an edge embedding, it needs to refine the cross association vectors across two graphs, whereas, we first compute the embeddings of query and corpus graphs separately and  independently of each other and then compute the alignment matrix. Thus, for our late models, the complexity of one forward pass for one pair of graphs is $O(K + |V\_q|^2 + |V\_c|^2)$ whereas for GMN, it is $O(K|V\_c|^2 + K|V\_q|^2)$, where $K$ is the maximum number of hops/layers in GNN architecture. The structure of our early interaction method XMCS and GMN are more comparable in terms of asymptotic complexity. However, because of regularities in our Gumbel-Sinkhorn computation, it was easier to tensorize XMCS, leading to a 3x speedup in experiments. Note that, even with such a speed up, XMCS outperforms GMN-match in terms of accuracy.
>
>
> > *How many nodes and edges are randomly added to the seed? What is the final size of the graph? This concern is of particular relevance given that exact MCS is NP-hard to compute, but the authors can generate the ground-truth MCS values “using the combinatorial formulations for exact MCCS and MCES,” raising the concern that whether the graph sizes are quite small so that even exact MCS computation is affordable. Figures 2 and 3 also show quite small graph pairs with ~20 nodes.*
>
> Here, we summarize the statistics for the number of nodes and edges in the graphs, for all the datasets.
>
>
> Dataset          |MM   |MR   |FM   |FR   |DD   |COX2 |MSRC
> -----------------|-----|-----|-----|-----|-----|-----|-----
> Avg, No. of nodes|12.48|12.5 |12.47|12.48|12.65|12.66|12.61
> Avg. No. of edges|13.05|13.22|13.19|13.17|17.97|12.88|17.58
>
>
>
> We elaborate more on the query/corpus generation process below.
> First, a set of 800 corpus graphs are generated using the randomized BFS traversal technique. Each corpus graph has node sets of size $|V|\in [10,15]$. Subsequently, we generate 500 seed query graphs, under the constraint that it should be subgraph isomorphic to a fraction $\eta \in [0.1,0.4]$  of the corpus graphs. This fraction is determined similar to prior works [Roy et al. [57], Lou et al. [39]. We then augment the seed query graphs with 1-2 nodes, with each of the new nodes having 3-5 edges. This larger seed graphs size (and correspondingly small augmentation), allows us to better control the range of MCS sizes across the set of corpus graphs. Given that the initial seed graphs had an MCS of 10 with $\eta \in [0.1,0.4]$  of the corpus graphs, and MCS<10 for the rest, this procedure lets us ensure a similar variation in MCS sizes across the corpus graphs, for any given query.
>
> Our choice of graph sizes was guided by  the needs of practical graph retrieval applications like molecular fingerprint detection, object detection in images and several prior work (Lou et al. [39], Roy et al. [57]). However, our method can easily scale beyond 20 nodes. See subsequent discussion.

---

> > ### Author Response · Authors · 2022-08-02
> > **Response to Reviewer UMf5 (Part 2)**
> >
> > > *Why would people use such approximate methods for MCS scores if the input graphs are small and the exact solvers can already handle them reasonably well? Without large graph experiments, the exact solver may be used as a baseline too, which is missing in the current experiments.*
> >
> >
> > Note that even for small graphs, MCS computation using combinatorial algorithms is challenging, esp. for 80K graph pairs. To elaborate, we present the time taken (in seconds) by our proposed models, and the combinatorial algorithm for 80 K graph pairs. Note that for the combinatorial method, we ensured the fastest possible inference times by allowing for maximum possible parallelization across all CPU cores.
> >
> >
> >  .    | LMCCS  | LMCES  | XMCS   | Combinatorial
> > ----------------------- | ------ | ------ | ------ | -------------
> > Inference time (in sec) | 13.776 | 28.129 | 31.101 | 290
> >
> >
> > We observe that the inference time of our slowest early interaction models is almost 10x faster than the combinatorial approach.  This is because even for such smaller sized graphs, combinatorial MCS computation is daunting with 80,000 graph pairs (100 query and 800 corpus). Larger graphs would make combinatorial “reference” methods completely impractical.
> >
> > It is true that we performed experiments on graphs of size < 20,  driven by the needs of practical graph retrieval applications like molecular fingerprint detection, object detection in images, etc. However, our method can easily scale beyond |V | = 20, as follows:
> >
> > Inference Time  (in sec) | Current Size | V = 30 | V =50 | V = 70 | V = 100
> > ------------------------ | ------------ | ------ | ----- | ------ | -------
> > LMCES                    | 0.036        | 0.048  | 0.052 | 0.071  | 0.106
> > LMCCS                    | 0.022        | 0.031  | 0.039 | 0.042  | 0.064
> > XMCS                     | 0.067        | 0.071  | 0.085 | 0.095  | 0.131
> >
> >
> > In the above table, the inference times are computed for 128 graph pairs with node set sizes $|V|\in [current size, 30,50,70,100]$. We observe that even with graph pairs for $|V| = 100$ nodes, the inference time is at most 3X of that of the time for current graphs. So the neural models can easily scale to larger graph sizes.
> >
> > > *A minor concern is the applicability of the proposed models, since the models are specifically designed for MCS metrics, more specifically, MCCS and MCES. I agree that MCS metrics are already important and worth specific models designed for them, but it would be interesting to discuss the potential issues if the proposed methods are used for other metrics, e.g. GED, or domain-specific similarity scores. This would make the contribution and the scope of usage clearer.*
> >
> > In this paper, we have demonstrated that the current models for computing GED (GOTSim [14], GMN [35],GEN [35]) and Subgraph-Isomorphism (NeuroMatch [39], IsoNet [57])  do not perform as well as MCS-specific models for the MCES/MCCS tasks. While currently there exist a variety of custom models for the different notions of graph similarity (GED/MCS/Subgraph-Isomorphism), it will be interesting to design an unifying model for all these scores in future work.

---

### Official Review · Reviewer_Q3Ju · 2022-07-12

**Rating:** 6
**Confidence:** 5
**Soundness:** 4 excellent
**Presentation:** 3 good
**Contribution:** 3 good

**Summary:**

In this work, the authors propose LMCES and LMCCS, which are late interaction neural architectures for MCES (counting the number of common edges) and MCCS (the maximum common connected subgraph) based graph retrieval respectively. Additionally the authors propose XMCS (cross-MCS) which is an early interaction model that works well for both MCES and MCCS based graph retrieval. The authors demonstrate the efficacy of their approach via empirical results.

**Questions:**

q1) why did you not include results related to to the training/inference times for the different baseline models along with the proposed approaches ?

**Limitations:**

The authors share multiple limitations of their work.

**Strengths And Weaknesses:**

Here are the main strengths of the current work :
1) The authors focus on graph similarity measurement which is a very important problem in the graph ML community.
2) The authors propose both early and late interaction models for determining the MCES and MCCS, via introducing a relaxation to the node alignment matrix between query and corpus graphs.
3) The authors demonstrate the efficacy of their approach via exhaustive empirical study using seven diverse datasets and six state-of-the-art late interaction baseline models.

Here are the main weaknesses of the current work :
1) The novelty factor of the proposed approaches is somewhat limited.
2) The authors should have included more results, including those related to the training/inference times for the different baseline models. Currently it is hard to compare whether the gain in performance is matched by the potential training/inference latency issues.

---

> ### Author Response · Authors · 2022-08-02
> **Response to Reviewer Q3Ju**
>
> > *The authors should have included more results, including those related to the training/inference times for the different baseline models. Currently it is hard to compare whether the gain in performance is matched by the potential training/inference latency issues.*
>
>
> We have compared the inference time of our method, against the most competitive (accuracy-wise) method GMN-match in the main submission. Table 4 summarizes the observations, which reveals that our method is significantly faster than GMN-match in terms of the inference time. Since in the context of information retrieval, inference time is much more important than training time, we only reported inference time in the main paper.
>
>
> Here, we report both the training and inference time (in seconds). The training time is computed for each batch of size 128 (which is the fixed hyperparameter used for the numbers reported in the paper). Inference time is computed  for the entire test set of 100 query graphs and 800 corpus graphs, with the maximum possible batch size allowed by our GPU - Nvidia TITAN X (Pascal).
> Method    | Training Time per batch  (in secs) | Inference time on test (in secs)
> ----------|-----------------------------------|---------------------------------
> GEN       |0.037                              |5.937
> SimGNN    |0.073                              |24.671
> GraphSim  |0.125                              |23.284
> NeuroMatch|0.027                              |6.532
> GOTSim    |0.259                              |72.590
> IsoNet    |0.069                              |13.956
> GMN       |0.426                              |99.542
> LMCES     |0.109                              |28.129
> LMCCS     |0.074                              |13.776
> XMCS      |0.159                              |31.101
>
>
> We observe that:
> (1) Among the late interaction models,  both LMCCS and LMCES are significantly faster than GOTSim. This is because GOTSim uses a combinatorial solver, which does not allow for batched processing and results in significant slowdown.
> (2) LMCES is comparable to GraphSim and SimGNN, in both training and inference times, while affording significantly better performance.
> (3) LMCCS is comparable to IsoNet in training and inference times, and is significantly faster than SimGNN, GraphSim and GOTSim.
> (4) GEN and NeuroMatch are significantly  faster than all late interaction models in terms of training and inference times. However, as shown in Table 1 of the main paper, both these models are outperformed by LMCCS,LMCES and IsoNet in most of the datasets.
> (5) Among the early interaction models,  XMCS is 3X  faster than GMN. The reason for this is explained in lines 338-344 in our main paper.

---

> > ### Comment · Reviewer_Q3Ju · 2022-08-08
> > **Thank you for your comments**
> >
> > Dear authors,
> > Thank you for your comments and feedback to different issues pointed by all reviewers. I would like to retain the original score I had assigned to this paper.

---

### Meta-Review · Area_Chair_6wnY · 2022-08-26

**Recommendation:** Accept
**Confidence:** Certain

**Metareview:**

This paper presents a neural method for the graph retrieval problem. Reviewers agree with the technical contribution of this paper, its empirical soundness, and the well written presentation. Discussions in rebuttal and additional experiments provided useful information and made this paper stronger. We suggest the authors update their next version according to reviewers’ suggestions and rebuttal discussions.

**Award:**

No

---

### Decision · Program_Chairs · 2022-09-14

Accept